# Landsat-8 Observations of Foam Coverage under Fetch-Limited Wave Development

**Vladimir A. Dulov \*** , **Ekaterina V. Skiba and Arseny A. Kubryakov**

Marine Hydrophysical Institute of Russian Academy of Sciences, 2 Kapitanskaya, 299011 Sevastopol, Russia; evskiba@mhi-ras.ru (E.V.S.); arskubr@yandex.ru (A.A.K.)
* Correspondence: dulov@mhi-ras.ru

**Abstract:** In this paper, we aimed to clarify the problem of foam coverage dependence on wave fetch, which is of interest in satellite microwave radiometry, but for which controversial results were reported previously. The classical approach to investigating developing waves was applied. That is, the waves are considered as coming from the coast under approximately constant wind velocity. The study includes two scenes of intensive katabatic winds in the Gulf of Lion and the Gulf of Tehuantepec. We used two Bands of Landsat OLI images to extract the wave spectral peak frequency and the sea fraction covered by foam simultaneously along the wave fetch. The distributions of the spectral peak frequency along the fetch obeying the classical wave growth law clearly showed that we observed the developing waves. Along the fetch, the sea surface covered with foam grows about three times with the power law. This development of foam coverage occurred at the range of dimensionless fetches from 50 up to 7000 if the fetch is scaled using wind velocity and gravity acceleration. A simple model of the foam coverage growth with wave fetch is suggested. We modeled wave energy dissipation rate using the JONSWAP wave spectrum for developing seas. The model explains the observations at the quantitative level. Reported results can be applied to investigations of tropical cyclones using satellite microwave radiometry.

**Keywords:** ocean; wind-driven waves; wave breaking; Landsat imagery; fetch-limited wave growth; sea foam; foam coverage modeling; storm winds

## 1. Introduction

Classical laws of wind-driven wave development are the basis of current numerical modeling and forecasting of sea surface waves [1–3]. Hence, they remain the subject of discussion, and both experimental [4–7] and theoretical [8–12] studies. The development of these laws results in parametric models of the evolution of dominant wind waves, which have an important practical application in studying and predicting wind wave fields generated by tropical cyclones [13–15].

Dissipation of wind waves emerges on the sea surface due to wave breaking. This supports permanent scientific interest in the investigation of wave breaking [16–22], related bubble fraction in water [23–28], and direct numerical modeling of these processes (see, e.g., [29–31]). Wave breaking intensifies the generation of air bubbles, water spray, and undersurface turbulent mixing, which amplifies exchange processes between the atmosphere and ocean [32–37]. Therefore, the problem of wave breaking evolution during wave development has direct application to both climate predictions [34,38–40] and modeling of tropical cyclone dynamics [34,41–43]. Together with wind velocity, wave development determines the variability of the wave field in the tropical cyclone characterized by varying wave ages from young to mature waves [13–15]. Thus, for tropical cyclone dynamics, the issue of how wave age impacts wave breaking and modulates air–sea interactions can be of principal significance.

Wave breaking through changing of sea surface physical properties affects the remote sensing signals and contributes to formation of typical images of tropical cyclones. The

challenging problem of tropical cyclone diagnostics via satellite data requires an understanding of the wave breaking field in the tropical cyclone. Tropical cyclones have typical images in the radio-brightness temperature of the sea [44,45]. Possible interpretation of these images can be related to wave breaking because, on the one hand, wave breaking influences the signal of the microwave radiometer [46–51], and, on the other hand, we can expect strong variations of the foam coverage in the tropical cyclone field [14].

Wave development from the coast obeys the power-laws [1–3]

$$\xi = \xi_0 \chi^{-q}, \tag{1}$$

$$\varepsilon = \varepsilon_0 \chi^p, \tag{2}$$

where

$$\varepsilon = Eg^2/U^4, \ \xi = f_p U/g, \ \chi = Xg/U^2 \tag{3}$$

are the dimensionless energy, spectral peak frequency, and fetch, respectively. $f_p$ is the spectral peak frequency, $X$ is the distance from the shoreline, $U$ is the wind speed, $g$ is the gravity acceleration. Wave energy $E$ is the variance of the sea surface displacement related to significant wave height,

$$H_S = 4\sqrt{E}.$$

Wave age is equal to

$$\zeta = c_p/U = 1/(2\pi\xi),$$

where

$$c_p = g/(2\pi f_p).$$

is the phase velocity of spectral peak waves in deep water. Coefficients $q, p, \xi_0$ and $\varepsilon_0$ in power laws were evaluated experimentally ([2,51–53] among others), but their clarification and possible dependence on wave age remain current scientific problems [5–9].

As waves develop under the summary action of wind energy input, of nonlinear energy transfer along the wave spectrum and dissipation of energy due to wave breaking [2,3], the study of wave breaking is of interest to understand the actual balance of these processes finally determining the form of wave development laws [8,9,12]. However, the dependence on the wave age of wave breaking intensity, particularly the surface fraction covered by wave breaking, $W$, remains unclear. For example, theoretical and experimental studies [20,54–57] reported that $W$ falls with the growth of $\zeta$. An analysis of the dataset obtained by various authors [58], field experiments [59–61], and satellite data [62] showed the inverse result. Additionally, field experiments [18,63] yielded non-monotonic dependence $W(\zeta)$. Among the possible causes of disagreements are the unsteadiness of wave fields [6,55] and influence of currents and tides [57,64] during wave breaking estimations. Wave development laws may be distorted for the unsteady wave field [6,7]. It is reflected in wave breaking intensity as reported in [55,57,60,64]. Thus, comparing the wave breaking estimations obtained in different time moments or experiments can lead to erroneous results if these effects are disregarded. However, if we perform wave measurements complemented with the wave breaking estimations simultaneously along the entire wave fetch, we can (i) inspect whether wave characteristics obey known stationary fetch-limited laws; (ii) if it does, consider wave breaking dependence on the fetch. Satellite data provides the opportunity to perform such experiments.

An evaluation of the parameters of the dominant wind waves was made from high-resolution optical images from Landsat-8, Sentinel-2a and 2b, WorldView-3, and other satellites [65–68]. Whitecaps on the crests of breaking waves and the spots of foam produced by bubbles ascending to the surface after the active phase of wave breaking can also be observed from satellites, see, e.g., [62,64,69,70]. The possibility to evaluate the $W$ using near IR images from Landsat-8 was demonstrated in the study [64]. In that study, the obtained dependence of $W$ on wind velocity agrees with previous measurements, and

responses of wave breaking to the atmosphere and ocean features quantitatively agree with known theoretical estimations (see list of references in [64]). Using the satellite image where the waves and their breaking are visible simultaneously, which covers an area of tens to hundreds of kilometers, we can investigate wave development augmented with new data on wave breaking that evolves as waves develop.

Figure 1 shows the QuikSCAT wind velocity field average from the years 2000 to 2009. Red rectangles in the figure show two near coastal areas, which are prominent due to a relatively high mean velocity caused by catabatic jets of Mistral and Tehuantepec winds. Waves generated by these winds develop from the shoreline in deep water conditions (see bathymetry of these areas in the figure). Wind wave investigations and wave modeling for these areas are presented in many papers (see, e.g., [55,63,71–73] for Mistral and [5,18,74] for Tehuantepec). Thus, these places are rather suitable for investigations of wave development laws. However, scenes free of clouds are difficult to find using Landsat optical images for the study. We analyzed Landsat-8 images for several years but could select only two scenes; these positions in the Gulf of Lion and Gulf of Tehuantepec are shown in zoomed maps in Figure 1.

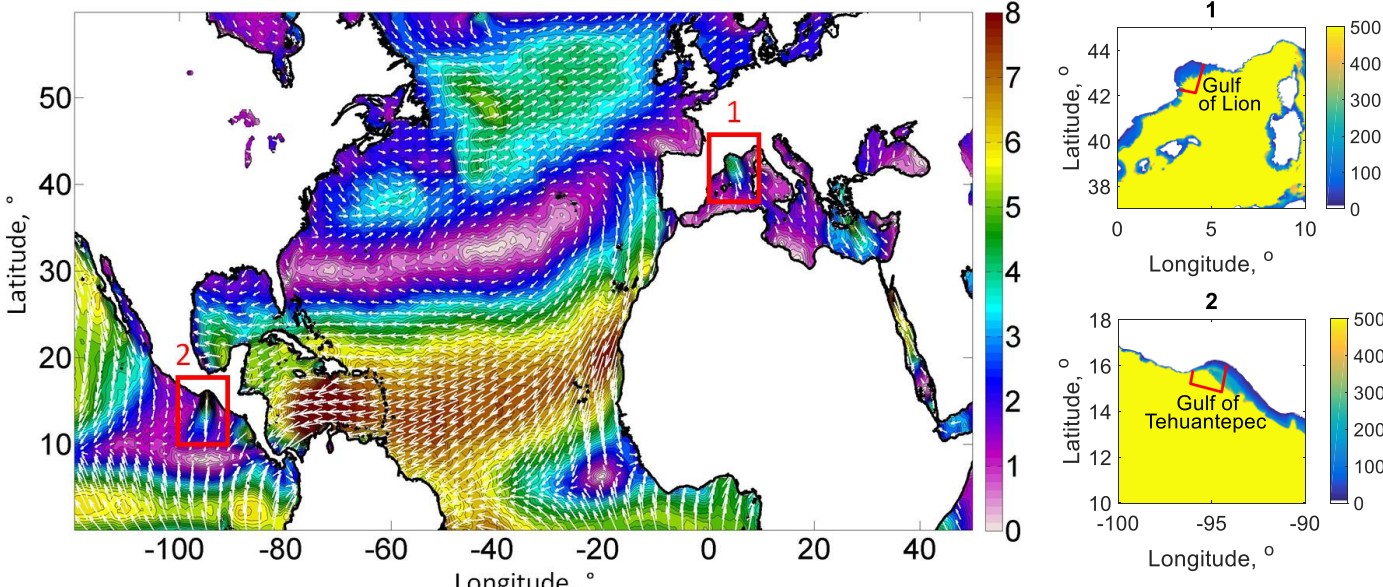

**Figure 1.** Average QuikSCAT wind velocity for 2000–2009. The color bar shows wind speed in m/s. Red rectangles show areas of Mistral winds (1) and Tehuantepec winds (2), in which zoomed maps are shown on the right. In zoomed maps, near coastal water depths are shown in m, and boundaries of Landsat-8 scenes are shown with red contours.

In this paper, the study of wave development in limited fetches was performed using optical images taken from Landsat-8, from which characteristics of both waves and their breaking were extracted. Two scenes are considered, wave growth under Mistral in the Gulf of Lion and wave growth under wind jet in the Gulf of Tehuantepec. Section 2 describes the data and methods used to evaluate the spectral peak frequency and surface fraction covered by wave breaking. Section 3 presents obtained wave development laws for the spectral peak frequency and surface fraction covered by wave breaking. Section 4 discusses obtained results and suggests their interpretation using a simple model. Section 5 resumes the article.

## 2. Data and Methods

### 2.1. Data Description

We used images from Landsat-8, obtained with radiometer OLI in visible and near IR bands with spatial resolution of 30 m. Data Level 1 were taken from the United States

Geological Survey (USGS) Global Visualization Viewer (http://glovis.usgs.gov/ accessed on 21 July 2021). For analysis, we selected two scenes in the regions of intensive katabatic winds, which form jets over the sea, expanding from tens up to hundreds of kilometers. The first scene was registered in the Gulf of Lion, the Mediterranean Sea, during the Mistral storm winds (10:49 GMT 5 March 2015), the second in the Gulf of Tehuantepec, Pacific, during the Tehuano storm winds (16:42 GMT 13 January 2015). We used ASCAT scatterometer data (Level 2B) of resolution 12.5 km as quasi-synchronous wind fields (10:22 GMT and 16:30 GMT, respectively), which correspond to wind velocity at a standard height of 10 m. These data were downloaded from the PODAAC archive, https://podaac.jpl.nasa.gov/dataset/ASCATA-L2-Coastal, accessed on 21 July 2021. The accuracy of wind measurements is 1.5 m/s in speed and 20° in direction [75]. Scatterometer data is unavailable near the shoreline (~15 km off the coast) due to the land contamination of the microwave signal. We estimated sea surface temperature from Landsat-8 TIRS measurements using a two-band algorithm proposed by Aleskerova et al. [76]. Data on near-surface air temperature were obtained from the European Center for Medium-Range Weather Forecasts ERA5 reanalysis. The meteorological conditions of the scenes are considered in Appendix A.

Figure 2 shows a fragment of the Landsat-8 image in the Gulf of Lion region in Bands 5 and 7 operating at 0.525–0.6 nm and 2100–2300 nm, respectively. In high winds, wave breaking is the main contributor to the sea brightness variations in the visible range [64]. It causes irregular patterns with high reflectance, see Figure 2a. In the near IR range, a wave breaking contribution is less significant, and dominant wind waves are well visible, see Figure 2b. Landsat-8 measurements have a higher signal-to-noise ratio than previous Landsats [77]. As a result, we can observe the brightness contrast of 0.006 sr$^{-1}$ between crests and troughs of waves, which are visible in Figure 2b. In this study, we determined the fraction of sea surface covered by wave breaking using the Band 5 data. The wind wave characteristics were estimated using the Band 7 data. Additionally, Band 5 and Band 9 data were used to filter out the land and clouds (see more detail about this data processing stage in [64]).

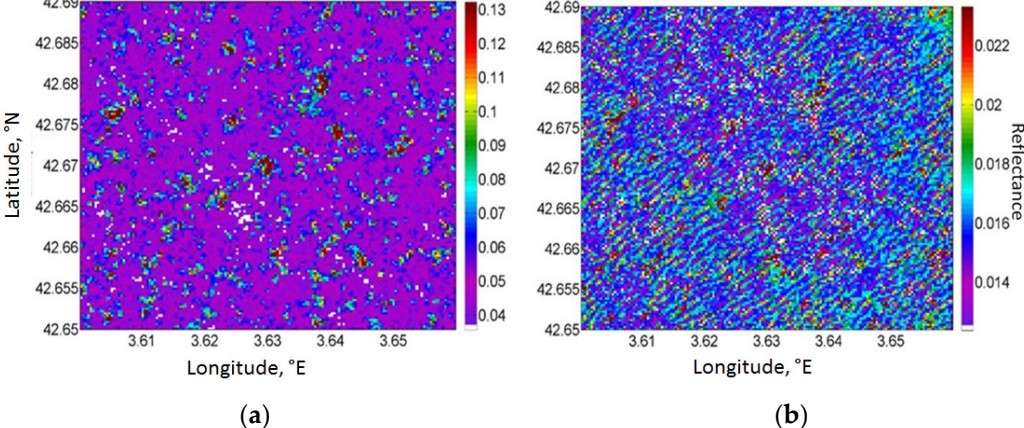

(**a**)　　　　　　　　　　　　　　　　　　　　　(**b**)

**Figure 2.** Fragments of the sea surface in Band 5 (**a**) and Band 7 (**b**) of Landsat-8 image. Reflectance is shown in sr$^{-1}$. The size of the fragment is about 4 km.

### 2.2. Method of Wave Measurements

Traditionally wave characteristics from images are measured using Fourier-transform and analyzing brightness spectra of image fragments, $S_B(k_x, k_y)$, where $k_x$ and $k_y$ are components of the wave vector (see, e.g., [65,68,78]). We estimated brightness spectra in the square fragments of 2 km × 2 km using a conventional method, see, e.g., [79]. Further, the sea elevation spectrum, $S_H$, can be obtained by dividing it by the modulation transfer function (MTF):

$$S_H = S_B / (G_x k_x + G_y k_y)^2$$

where $G_x$ and $G_y$ are constants, the determination of which requires a special calibration. For calibration, an analysis of satellite sun glitter imagery of the ocean [65], an iteration matching of brightness field and their physical model [68], and stereo processing of sea images [78] were used. If MTF is unknown, neither the energy scaling factor nor angular distribution can be evaluated without significant distortions [78]. Therefore, in this paper, we used the only robustly quantified feature of the brightness spectrum, the wavenumber of the spectral peak $k_p = (k_{px}^2 + k_{py}^2)^{1/2}$. Then, for wavelengths from 60 m to 120 m, wavelength errors are from 2.5 m to 10 m (see, e.g., [79]). We also considered peak value $S_{\max} = S_B(k_{px}, k_{py})$ to speculate on wave energy at the quality level.

### 2.3. Method of Wave Breaking Measurements

In the paper [64], an algorithm for quantitative evaluation of the sea surface fraction covered with bright whitecaps and foam spots was suggested (referenced further as the KKS algorithm). The method of the KKS is based on the fact that the reflectance of wave breaking area and clear water surface differ by the order of value [80]. The KKS algorithm allows estimation of the dependence of $W$ on wind velocity, which agrees with other authors' measurements [20,48,64]. Responses of wave breaking to the atmospheric and oceanic features estimated with the KKS algorithm quantitatively agree with known theoretical estimations (see list of references in [64]). In this study, we applied the KKS algorithm for wave breaking measurements. The $W$ was estimated in each pixel of the image (a square of 30 m $\times$ 30 m) which does not locate to the near shoreline area and areas contaminated by clouds. This approach does not resolve the spatial scale of wave breaking. However, it is based on the average reflectance to which all breaking waves contribute. Thus, we considered breaking of waves from spectral peak up to about 1 m wavelength, which generates a water-bubble mixture with high reflectance [19,21]. These measurements differ from traditional wave breaking estimations based on sophisticated analyses of optical sea images [18,19,81]. They are closer to radiometric estimations of foam coverage [50,62].

## 3. Results

### 3.1. Approach to Data Analysis

Figures 3 and 4 show the source data of our study: the wind speed, $U$, and wind direction at the height of 10 m, the fraction of sea surface coved with whitecaps, $W$, the wavelength of spectral peak waves, $\lambda_p = 2\pi/k_p$, the values of spectral maximum, $S_{\max}$, water, and air temperatures for two considered scenes. Empty pixels in the maps correspond to areas excluded from analysis due to cloud impact or proximity to the shoreline. As the spatial resolution of the image was 30 m, we obtained only wavelengths of dominant waves longer than 60 m. Thus, wavelength measurements are absent in areas close to the shoreline, where the length of developing waves is less.

As follows from Figures 3 and 4, offshore wind jets over the sea expanding several tens of kilometers are observable in both scenes. Waves develop along the jets. This is indicated by the growth of spectral peak wavelength and wave energy, which at the qualitative level emerges as the growth of the spectral maximum. Wave development is accompanied by the intensification of wave breaking. Both events occurred at approximately constant wind velocity. In Figure 2b atmospheric internal lee waves in the Gulf of Lion can be observed. White-capping manifests these as periodic strips of wave breaking intensification. See a more detailed description of this phenomenon in [64].

It is known that a scatterometry wind velocity does not match the in-situ value exactly [60,75]. However, although the scatterometer winds are of the highest possible resolution, this is still insufficient to capture small-scale wind variability, which is resolved in the maps of the Landsat whitecap fraction [64]. In our study, a comparison of Figure 3a,b shows that we see no indication of atmospheric lee waves in Figure 3a, but they are visible in Figure 3b. Therefore, we further analyzed the data relying on patterns of whitecap spatial distributions.

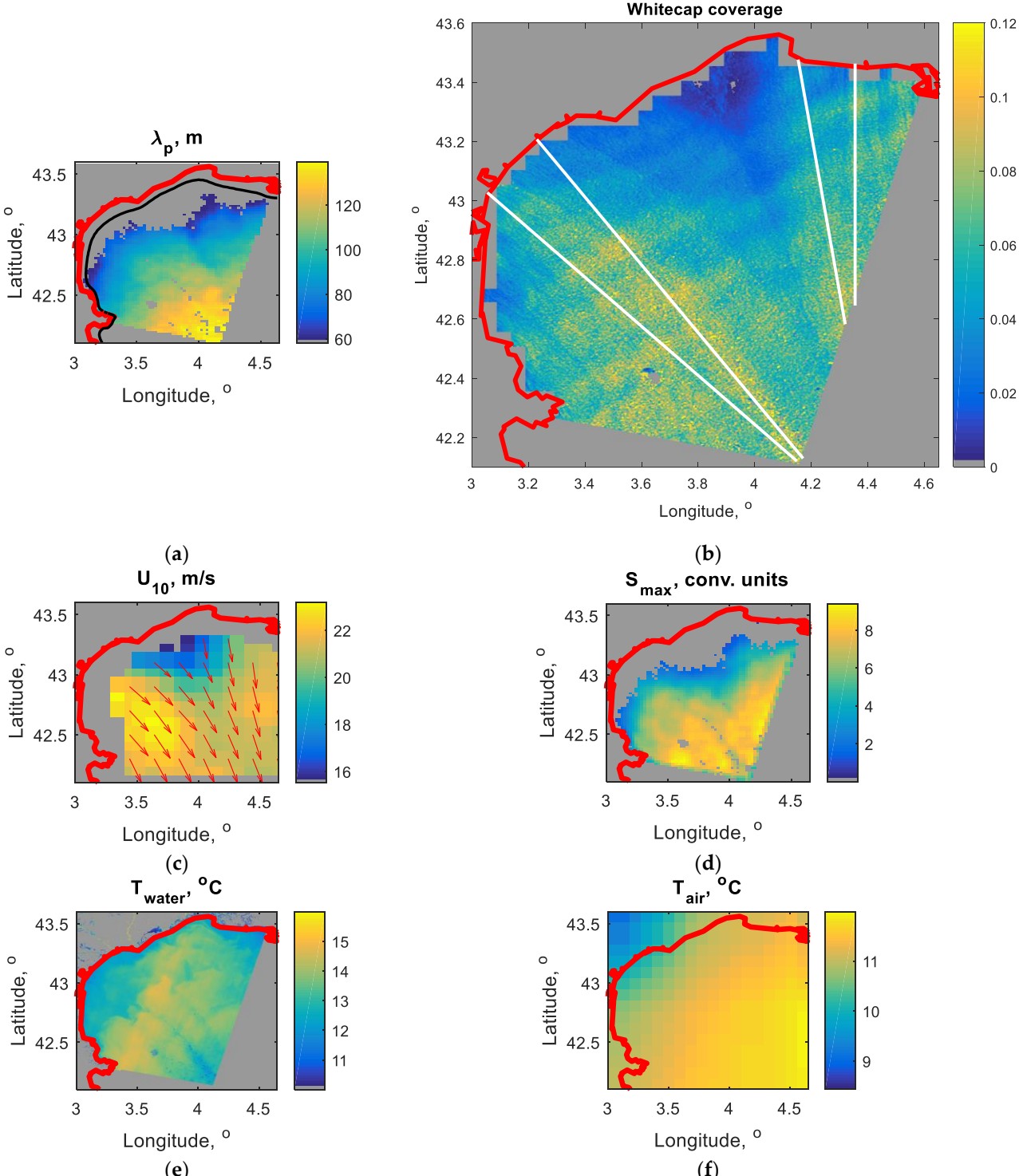

**Figure 3.** Wave development in the Gulf of Lion: (**a**) wind velocity vectors; (**b**) fraction of sea surface coved with wave breaking, *W*; (**c**) wavelength of spectral peak waves, $\lambda_p$; (**d**) the values of spectral maximum, $S_{max}$; (**e**) water temperature; (**f**) air temperature. The thick red line shows the shore contour. Locations of analyzed sections are shown in the plate (**b**) using white lines. The black line in plate (**c**) shows 30 m-isobath.

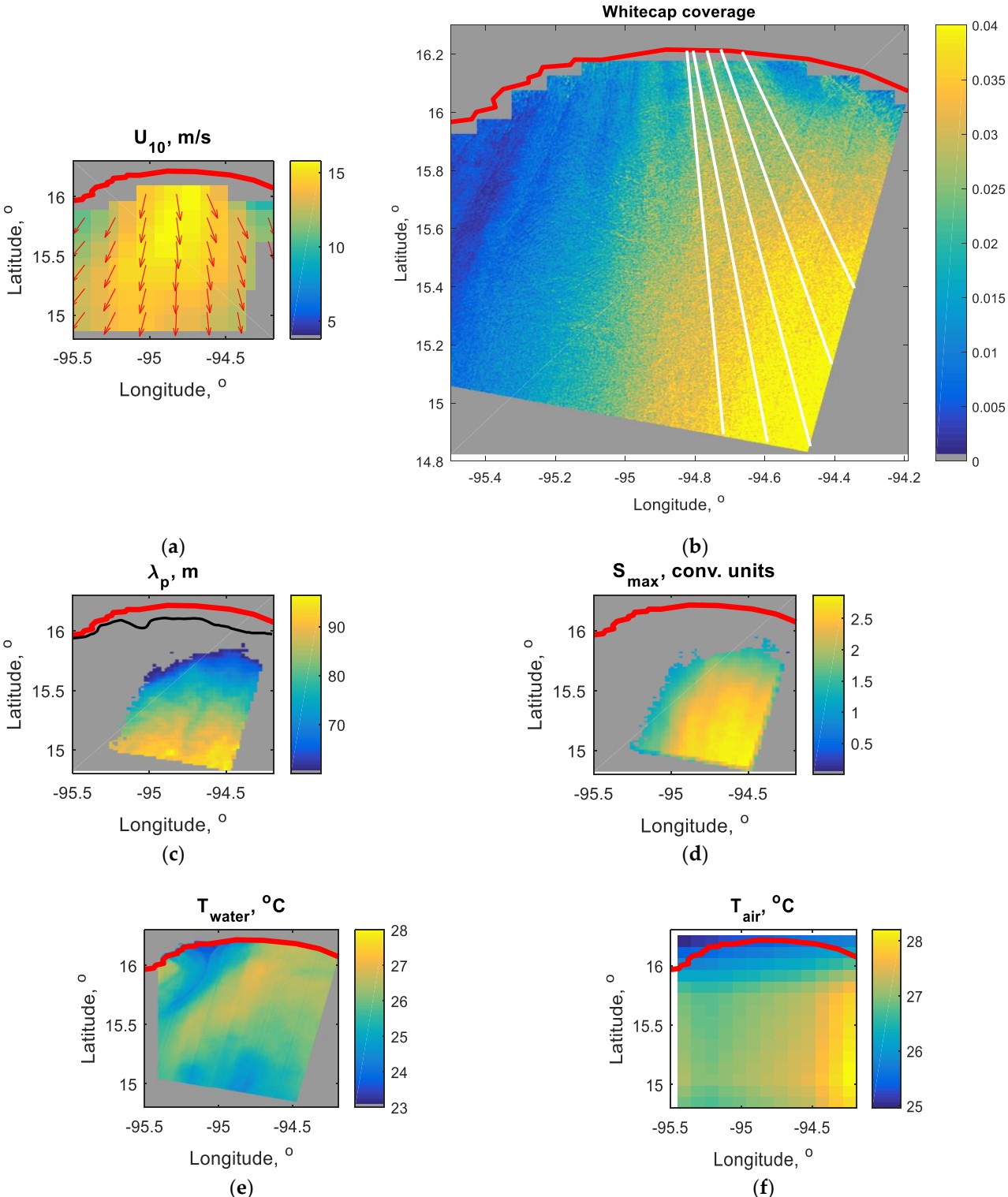

**Figure 4.** Wave development in the Gulf of Tehuantepec: (**a**) wind velocity vectors; (**b**) fraction of sea surface coved with wave breaking, *W*; (**c**) wavelength of spectral peak waves, $\lambda_p$; (**d**) the values of spectral maximum, $S_{\max}$; (**e**) water temperature; (**f**) air temperature. The thick red line shows the shore contour. Locations of analyzed sections are shown in the plate (**b**) using white lines. The black line in plate (**c**) shows 30 m-isobath.

Figures 3b and 4b show that the wind blow splits into several thin jets. It is more evident in Figure 4b, where no atmospheric internal waves are presented. It can be ex-

pected that wave development along such jets is the closest to classical ideas (1) and (2) on wave growth from the shore. Using visual inspection of $W$ fields, we selected sections matching thin wind jets and crossing the areas of wavelength measurements. We selected four sections in the Gulf of Lion (see Figure 3b) and five sections in the Gulf of Tehuantepec (see Figure 4b). Further analysis was conducted only for these sections. Although wave development laws imply the wind blowing perpendicular to the shoreline, Ardhuin et al. [4] argued that slanting fetch does not affect wave development, at least when wave direction relative to the normal to the shoreline does not exceed 30 degrees. As classical representations (1) and (2) imply the spatially uniform wind velocity [1,2], the wind speed is commonly averaged to be used in wave development analysis [5,7,53]. The wind velocities averaged over all sections are $U_L = 21.3$ m/s and $U_T = 14.4$ m/s for the Gulf of Lion and the Gulf of Tehuantepec, respectively. Further, these constant values were used to analyze wave and whitecap development.

We obtained 201 and 206 estimations of spectral peak wavelength along sections in the Gulf of Lion and the Gulf of Tehuantepec, respectively. The source whitecap fraction $W$ estimated with a spatial resolution of 30 m has high random variations. We averaged $W$ along the section over 2 km intervals corresponding to the resolution of wavelength estimations $\lambda_p$. Additionally, the $W$ was measured in near shoreline areas where the wavelengths were shorter than 60 m and waves were unresolvable. As a result, we have 267 and 331 estimations of whitecap coverage $W$ in the Gulf of Lion and the Gulf of Tehuantepec, respectively.

### 3.2. Spectral Peak Frequency versus Fetch

The wave fetches $X$ for points of each section were determined as a distance from the point to the shoreline along the section. Spectral peak frequency was calculated from spectral peak wavelength via the linear dispersion relation for deep water,

$$f_p = \left( \frac{g}{2\pi \lambda_p} \right)^{1/2}.$$

Dimensionless variables $\xi = f_p U / g$ and $\chi = X g / U^2$ were obtained using averaged wind velocities. Classical laws (1) and (2) are valid for deep water only [1]. The waves are considered to be in deep water if half of the wavelength does not exceed the sea depth [3]. Figures 3c and 4c show 30 m-isobaths to confirm that our wavelength measurements were in deep water areas.

Figure 5 shows the obtained law of wave development for both scenes together with data from [52] (see Table 1) and [7] (see Table A1), obtained by in situ measurements in the Black Sea, and [5] (see Tables 1 and 2), obtained by aircraft measurements in the Gulf of Tehuantepec. All data agree, and the clouds of our data are densely localized and overlap. Least-square estimation applied to logarithms of our data yields the power law

$$\xi_1 = \xi_{01} \chi^{-q_1}, \ \xi_{01} = 1.45^{+0.04}_{-0.04}, \ q_1 = 0.232 \pm 0.004 \tag{4}$$

With a coefficient of determination $R^2 = 0.93$. Hereinafter, the confidence intervals correspond to double standard error [79]. The joint cloud of all data in the figure leads to the power law

$$\xi_2 = \xi_{02} \chi^{-q_2}, \ \xi_{02} = 2.04^{+0.08}_{-0.07}, \ q_2 = 0.271 \pm 0.005 \tag{5}$$

with a coefficient of determination $R^2 = 0.96$. Figure 5 shows both laws. Both estimations agree with generally accepted views on the values of parameters of the law (1) (see, e.g., [6,8,9]).

Thus, two selected scenes can be, considered with certainty as wave development at limited fetches, and data on dimensionless fetch $\chi$ can be used to study the law of whitecap coverage development. Furthermore, in this paper, we accept the law (4) as based on a more extensive dataset and have a better coefficient of determination.

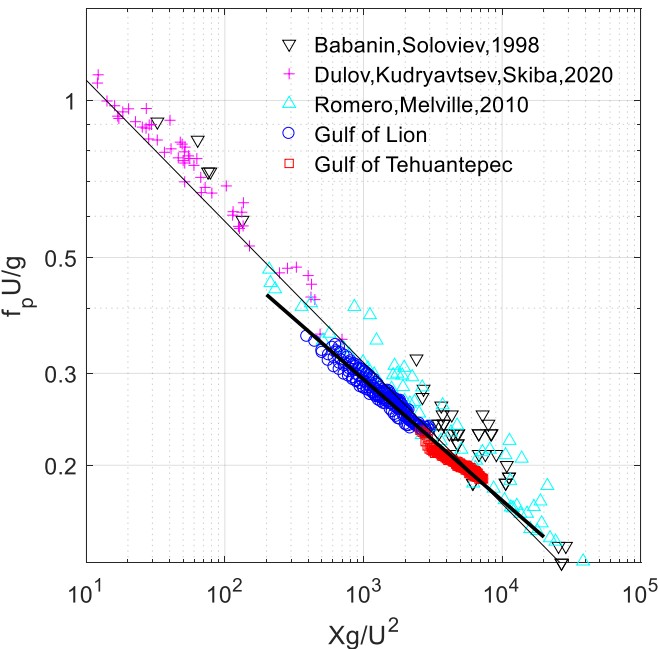

**Figure 5.** Dependence of dimensionless spectral peak frequency on dimensionless wave fetch. Thick and thin lines show power laws (3) and (4), respectively.

### 3.3. Whitecap Coverage versus Fetch

Figure 6a shows whitecap coverage *W* in dependence on the distance *X* from the shoreline. The values of *W* certainly grow with the fetch increasing about three times. For the Gulf of Tehuantepec data, it is not caused by a local increase in wind speed, see Figure 4, where wind speed decreases with distance from the shore. As argued in Appendix A, the growth of *W* cannot be explained by changes along the sections in wind velocity, water, and air temperatures, or atmosphere stratification]. Figure 6b shows the same data versus wind velocity compared with *W(U)* curves from other studies. Our data are presented as bin-averaged over dimensionless fetch *χ*.

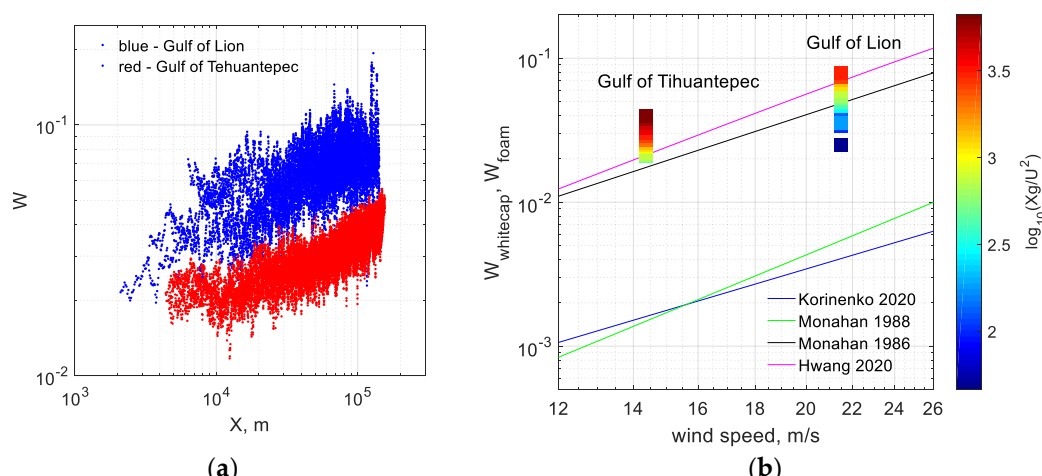

(**a**)            (**b**)

**Figure 6.** Whitecap coverage dependence on (**a**) the fetch and (**b**) wind velocity. In plate (**b**), these study data are color-coded using dimensionless fetch.

At a given wind velocity, the values of whitecap coverage from various studies vary more than the order of the value [48] (Figure 1), [20] (Figure 1). This data spread is mainly related to the fact that the wave breaking act consists of active phase A with a bright

whitecap on the crest of a breaking wave and passive phase B, which is visible due to the gradual ascending of the plume of air bubbles after wave breaking [24,58,81,82]. Phase B is commonly referenced as foam. Whitecap coverage belonging to phase A is significantly less than the foam coverage (see, e.g., [83]). Methods of whitecap coverage measurements based on brightness thresholding lead to data where both phases are mixed to varying degrees. Figure 5 aims to show to which phase our data belong. It shows the upper and lower boundaries of the parameterizations cloud from the paper [48] corresponding to the active wave breaking phase A, $W_{whitecap}$, and total surface fraction covered by both phases, where phase B dominates, $W_{foam}$:

$$\left(W_{whitecap}\right)_M = 2.92 \cdot 10^{-7} U^{3.204} \tag{6}$$

$$\left(W_{whitecap}\right)_K = 3.5 \cdot 10^{-6} U^{2.3} \tag{7}$$

$$\left(W_{foam}\right)_M = 1.95 \cdot 10^{-5} U^{2.55} \tag{8}$$

$$\left(W_{foam}\right)_H = \begin{cases} 0.3(u_* - 0.11)^3 & 0.11 < u_* < 0.4 \\ 0.07u_*^{2.5} & u_* \geq 0.4 \end{cases} \tag{9}$$
$$u_* = C_D^{1/2}U, \ C_D = 10^{-4}(-0.016U^2 + 0.967U + 9.058)$$

Equations (6) and (8) are obtained in [84,85], respectively. They agree with more recent results (7) and (9) obtained in [21,50], respectively. Equation (7) was derived using video processing accounting for whitecap kinematics to filter out slowly moving foam [81]. Equation (9) summarizes whitecap coverage estimations derived from microwave radiometer measurements and accounts well for sea foam contribution. The friction velocity $u_*$ in (9) is calculated using the drag coefficient $C_D$ presented in [50]. Figure 5 demonstrates that our data belong to phase B growing with the increase in wave fetch.

Figure 7 illustrates the wave development analysis of the foam coverage data. As supported by Figure 7a, the data can be described by the equation

$$W_{foam} = A_1 U^n \chi^{b_1}, \ A_1 = \left(8.83^{+2.79}_{-2.12}\right) \cdot 10^{-6}, \ n = 2.40 \pm 0.07, \ b_1 = 0.244 \pm 0.014 \tag{10}$$

with a coefficient of determination $R^2 = 0.91$. Coefficients in (10) were least square-approximated using the logarithms of the data. The obtained value of $n$ falls into the range of the estimations known in the literature [20,48,58]. Using (5), we come to dependencies of $W_{foam}$ on dimensionless spectral peak frequency and wave age:

$$W_{foam} = 1.6 \cdot 10^{-5} U^n \xi^{-0.83}, \tag{11}$$

$$W_{foam} = 7.3 \cdot 10^{-5} U^n \zeta^{0.83}. \tag{12}$$

Equations (10)–(12) are derived for data from the ranges of $50 < \chi < 7 \cdot 10^3, 0.19 < \xi < 0.35$, and $0.45 < \zeta < 0.84$, respectively. Figure 7b shows the degree of compliance with the data and parameterization (12). Equations (10)–(12) provide laws of development for sea surface fraction covered with foam under limited wave fetches.

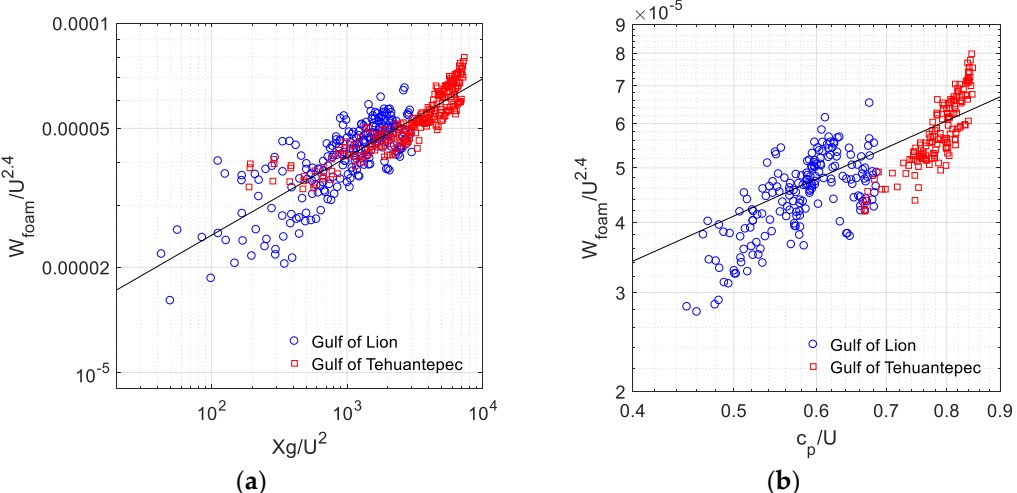

**Figure 7.** Laws of development for sea surface fraction covered with foam under limited wave fetches as dependencies of $W_{foam}$ on (**a**) dimensionless fetch and (**b**) wave age. Black lines show Equation (10) (**a**) and Equation (12) (**b**).

## 4. Discussion

Experimental and theoretical studies by [20,54–57] reported that $W$ falls with an increase in wave age $\zeta$. Non-monotonic dependence $W(\zeta)$ was found in field experiments [18,63]. On the other hand, field experiments of Stramska and Petelski [59], Goddijn-Murphy et al. [60], and Sugihara et al. [61] indicated that at a given wind velocity, the $W$ is higher for more developed rather than young waves. Analysis of an enormous volume of satellite data [60] showed that at given wind velocities exceeding 12 m/s, the $W_{foam}$ grows with the increase in wave height and period of dominant waves (see right columns in Figures 6 and 7 in their article). This indirectly supports the increase in $W_{foam}$ as waves develop. Bortkovskii and Novak [58] analyzed a combined dataset of many authors to reveal possible relations of $W$ and various dimensionless combinations of wind velocity, wave parameters, and kinematical viscosity of water $\nu$. We note the following dependence derived by them (see string 13 in Table 2 in their article):

$$W_{foam} \sim \left( u_*(\nu\, f_p)^{-1/2} \right)^{2.1}. \tag{13}$$

As follows from (9), the drag coefficient $C_D$ equals 0.0020 and 0.0022 at wind velocities of 14.4 and 21.3 m/s, respectively. So further, we consider the $C_D$ constant. Then Equation (13) means

$$W_{foam} \sim \nu^{-1/2} U^{3.15} \zeta^{-1.05}, \tag{14}$$

that is, the $W_{foam}$ grows if waves develop at a constant wind velocity.

As the growth of $W_{foam}$ with an increase in wave age was found in our study, we suggest an interpretation of this result. Following [86–89], we consider the $W_{foam}$ is proportional to wave energy dissipation rate due to wave breaking:

$$W_{foam} \sim \int_0^\infty D(f)df.$$

In this approximate consideration, we omit the angle dependency of the integrand. The spectral rate of energy dissipation $D(f)$ can be evaluated through energy input to waves from the wind [90,91]:

$$D(f) \approx \beta(f)S(f),$$

where $\beta$ is the wind growth rate, and $S(f)$ is the frequency wave spectrum. To model wave development, we adopt the spectrum parameterization JONSWAP [2] depending on dimensionless wave fetch $\chi$:

$$S(f) = \alpha g^2 (2\pi)^{-4} f_p^{-5} F(\eta), \ \eta = f/f_p, \ \alpha = 0.076\chi^{-0.22}.$$

The wind growth rate we take in the form derived theoretically to describe wave development laws (1) and (2) [92]:

$$\beta(f) = 0.05 \frac{\rho_a}{\rho_w} 2\pi \ f \left( \frac{2\pi f U_{10}}{g} \right)^{4/3}, \tag{15}$$

where the ratio of air and water densities is $\rho_a/\rho_w = 1.3 \cdot 10^{-3}$. In studies [10,11], numerical simulations of wave development were performed for various parameterizations of the $\beta(f)$, and the Form (15) was found the best consistent with experimental, theoretical, and numerical considerations. So, the foam coverage equation takes the form

$$W_{foam} \sim 0.05\alpha \frac{\rho_a}{\rho_w} (2\pi)^{-5/3} g^{2/3} U^{4/3} f_p^{-5/3} \int_0^\infty \eta^{7/3} F(\eta) d\eta,$$

and finally,

$$W_{foam} = A\alpha(\chi) U^3 \xi^{-5/3} = A(2\pi)^{5/3} \alpha(\chi) U^3 \zeta^{5/3}, \ A = const. \tag{16}$$

Figure 8 summarizes the results of the discussion. First, it shows the uncertainty of known data regarding the dependency of wave breaking coverage on fetch and wave age. Data [18,55,58] are presented as examples. Bin-averaged data of Kleiss and Melville were taken from Figure 9b,d of [18]. Parameterizations of Lafon et al. ([55], Equation (12)) and Callaghan et al. ([57], Equation (3)) are

$$W_L = 0.106\chi^{-0.24},$$

$$W_C = 0.000311\zeta^{-4.63},$$

respectively. These results are obtained by combining data for different wind velocities, so they are independent of $U$, although such a dependence should be [62].

Second, Figure 8a,b show our parameterizations (10) and (12), respectively, in the ranges of parameters in which they are derived in this study. Note that disagreement in values with other data may be linked with different methods of wave breaking measurements. Additionally, the slope of $W_{foam}$ dependency (13) on wave age is shown in Figure 8b. Model estimations (16) are shown for wind speeds of 14.4 and 21.3 m/s with the constant $A = 0.65 \cdot 10^{-4} (\text{m/s})^{-3}$, selected to locate the curves in the considered ranges of $W_{foam}$.

As follows from Figure 8, the Model (16) explains well the data obtained in this study. Additionally, Equation (13), following from the analysis of Bortkovskii and Novak [58], shows a close tendency. Thus, the data considered can be physically interpreted in the frame of general laws of wave development. However, it will be interesting to find their interpretation considering the dynamics of bubbles generated in wave breaking. Achievements in laboratory research of bubble fraction (see, e.g., [23–28] among others), and also direct numerical modeling of relevant processes (see, e.g., [29–31]) give the hope that more accurate explanation of results (10)–(12) will be found in the future.

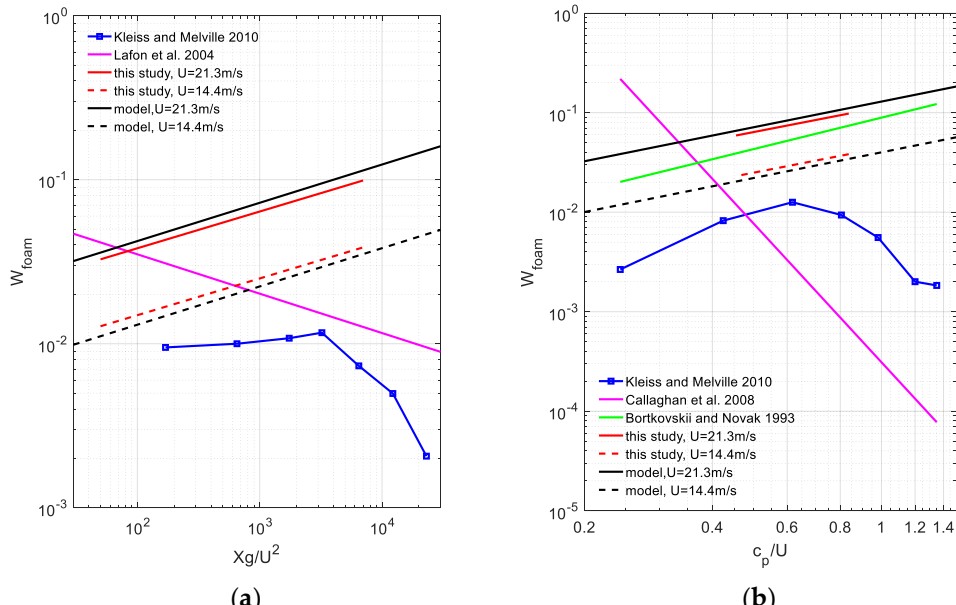

**Figure 8.** Comparison of experimental and model results from this study with data [18,55,57,58] for dependency of $W_{foam}$ on (**a**) wave fetch and (**b**) wave age. Model lines obey Equation (16).

## 5. Conclusions

This paper reports an investigation of wave growth laws under limiting fetches focusing on the wave spectral peak frequency and sea foam coverage due to wave breaking and using Landsat-8 imagery. Two scenes were selected with intensive katabatic winds in the Gulf of Lion and the Gulf of Tehuantepec. The spectral peak frequency and the sea foam coverage were extracted from Band 7 and Band 5, respectively, of the same Landsat OLI images at the various fetches along wind jets. We aimed to solve the problem of foam coverage dependence on wave fetch, for which controversial results were reported previously. To our knowledge, a one-moment-registered distribution of developing waves along the fetch augmented with foam coverage estimations was investigated for the first time.

As found, the distribution of the spectral peak frequency along the fetch obeys classical wave growth law and agrees well with previously obtained experimental results [5,7,52]. We observed the developing waves under approximately constant wind velocity at dimensionless fetches of a wide range from 50 to 7000. The distribution of the foam coverage shows approximately three-times the growth with a fetch increase, obeying power-law dependencies on fetch, spectral peak frequency, and wave age, see Equations (10)–(12). It supports the findings of Stramska and Petelski [59], Sugihara et al. [61], and Salisbury et al. [62], that foam coverage is higher for older than for younger waves. At the qualitative level, the lows (11) and (12) are close to Equation (13) obtained by Bortkovskii and Novak [58]. A simple model of the foam coverage growth with wave fetch is suggested using the JONSWAP [2] wave spectrum for developing seas. The model explains the observations at the quantitative level.

We anticipate that reported results on the significant growth of the foam coverage under wave development can be further used in different research applications, in particular for the satellite-based investigations of tropical cyclones using microwave radiometers, where the foam contributes to radio emission anomalies [44,49,93].

**Author Contributions:** Conceptualization, V.A.D.; methodology, V.A.D. and A.A.K.; software, A.A.K., E.V.S. and V.A.D.; formal analysis, V.A.D. and A.A.K.; investigation, A.A.K.; data curation, E.V.S.; writing—original draft preparation, V.A.D. and E.V.S.; writing—review and editing, V.A.D. and A.A.K. All authors have read and agreed to the published version of the manuscript.

**Funding:** This work was funded by the Russian Science Foundation Grant No. 21-17-00236 (https://rscf.ru/en/project/21-17-00236/, accessed on 14 April 2023).

**Data Availability Statement:** The data used in this study are openly available in the United States Geological Survey (USGS) Global Visualisation Viewer (http://glovis.usgs.gov/ accessed on 21 July 2021) and the PODAAC archive (https://podaac.jpl.nasa.gov/dataset/ASCATA-L2-Coastal, accessed on 21 July 2021).

**Acknowledgments:** The authors are grateful to Professor Vladimir Kudryavtsev (Russian State Hydrometeorological University, Saint-Petersburg, Russia) for useful discussions. This study was performed in the frame of the Russian Science Foundation Grant No. 21-17-00236 using information and computing resources from the Marine Hydrophysical Institute of the Russian Academy of Sciences within the State Assignments FNNN-2021-0002 and FNNN-2021-0004.

**Conflicts of Interest:** The authors declare no conflict of interest.

## Appendix A. Meteorological Conditions

The dependency of foam coverage $W_{foam}$ on water temperature $T_W$, air temperature $T_A$, and stratification of the near water layer of the atmosphere, characterized by the difference $\Delta T = T_A - T_W$, was revealed in several studies (see, e.g., [58,60,76,82]). Further, we consider whether the variations of these characteristics and wind velocities along analyzed sections explain observable variations of *W* depicted in Figure 5a.

Figure A1 summarizes the variability of wind velocity and water and air temperatures presented in Figures 2 and 3. These parameters are bin-averaged over the distance from the shoreline along all the sections. Table A1 gives the minimum and maximum values of parameters on the sections and expected relative responses $\Delta W_{foam}/\overline{W}_{foam}$ on such variations. Response to wind velocity variations was estimated adapting that $W_{foam} \sim U^3$:

$$\Delta W_{foam}/\overline{W}_{foam} \approx 3\Delta U/\overline{U}.$$

In a study by Salisbury et al. [60], an analysis was performed of giant data volume on $W_{foam}$ retrieved from satellite-based radiometric observations at a frequency of 37 GHz ($W_{37}$). In Figures 9b and 10b,d of their article, the responses $\Delta W_{foam}/\overline{W}_{foam}$ are shown on $T_W$, $T_A$, and $\Delta T$, respectively. They derived these estimations for various wind velocities up to 20 m/s. We used these Figures for wind velocities of 20 m/s and 14.3 m/s for filling Table A1.

**Table A1.** Estimations of relative responses $\Delta W_{foam}/\overline{W}_{foam}$ on variations of wind velocity $U$, water temperature $T_W$, air temperature $T_A$, and atmosphere stratification characterized by $\Delta T = T_A - T_W$.

| | Gulf of Lion | | | Gulf of Tehuantepec | | |
|---|---|---|---|---|---|---|
| | min | max | $\left\lvert\Delta W_{foam}\right\rvert/\overline{W}_{foam}$ | min | max | $\left\lvert\Delta W_{foam}\right\rvert/\overline{W}_{foam}$ |
| $U$, m/s | 19.2 | 22.1 | 0.4 | 13.0 | 15.5 | 0.5 |
| $T_W$, °C | 10.9 | 13.8 | <0.01 | 25.1 | 27.5 | <0.1 |
| $T_A$, °C | 10.7 | 11.7 | <0.02 | 25.5 | 27.5 | <0.1 |
| $\Delta T$, °C | −2.3 | −0.2 | <0.04 | −0.4 | 2.2 | 0.07 |

As follows from Table A1, the variability of meteorological parameters along the sections cannot explain the increase in foam coverage of 3.6 times in the Gulf of Lion and 2.7 times in the Gulf of Tehuantepec, demonstrated in Figures 5 and 6.

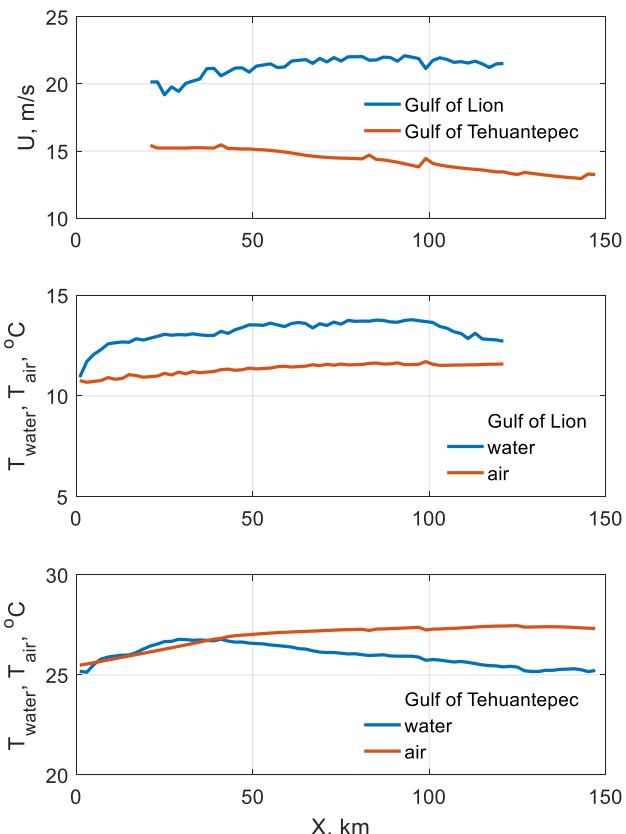

**Figure A1.** Wind velocity for two scenes (top), air and water temperatures in the Gulf of Lion (middle), and air and water temperatures in the Gulf of Tehuantepec (bottom) bin-averaged along the distance from the coast.

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
