# Peer review of "Landsat-8 Observations of Foam Coverage under Fetch-Limited Wave Development"

_remotesensing, doi:10.3390/rs15092222_

Round 1

Reviewer 1 Report

The authors investigate sea foam coverage due to wave breaking in two gulfs using Band 7 and Band 5 of Landsat 8 images. The fetch-limited foam coverage is obtained by data-fitting and expressed by a power-law in term of spectral peak frequency and fetch. Comparison of the expressions with other works shows fair agreement. Based on the innovation of the research method and the reliability of the results, this draft can be recommended for acceptance after minor revision. I have some comments for authors’ responses or additions.

1.     The wavelengths are about from 60 to 120m estimated from peak brightness spectrum for a 2 km x 2 km fragment.  The spatial resolution of Landsat 8 imagery is 30 m. Can the authors evaluate the error in peak wave number estimate due to image resolution?

2.     Equation (5) and (7) proposed by other authors. It is suggested to modify the symbols for Wwhitecap and Wfoam in (5) and (7) to be different from that of (6) and (8), respectively.

3.     The authors propose the expression for Wwhitecap and Wfoam by (6), (9), (10) and (11) in term of U, χ, ξ and ζ. Only Wfoam in (8) are expressed in term of CD and u* which are related to U. Is it possible to express Wfoam by U instead of u*?

4.     Coefficient of determination (R2) is commonly used to show the performance of model predictions.  It is suggested to show R2 between the observations and the predictions.

Author Response

We thank the reviewers for their thoughtful comments on our manuscript. Following reviews, we made substantial modifications to improve the original manuscript. All of the reviewers’ comments have been taken into account.

Bellow original Reviewer’s comments are italicized, 10 pt., and our responses are written in “normal” print, 12 pt. 

Response to Reviewer #1 comments

  1. The wavelengths are about from 60 to 120m estimated from peak brightness spectrum for a 2 km x 2 km fragment.  The spatial resolution of Landsat 8 imagery is 30 m. Can the authors evaluate the error in peak wave number estimate due to image resolution?

Response: The error in peak wave number estimate is 2π/2000m=0.0031rad/m, regardless of the spatial resolution of the image. The spatial resolution determines the maximum wavenumber we can get from the image. We added in the text (lines 176-177): “Then for wavelengths from 60 m to 120 m, wavelength errors are from 2.5 m to 10 m (see, e.g., [73]).”

  1. Equation (5) and (7) proposed by other authors. It is suggested to modify the symbols for Wwhitecapand Wfoamin (5) and (7) to be different from that of (6) and (8), respectively.

Response: Corrected

  1. The authors propose the expression for Wwhitecapand Wfoamby (6), (9), (10) and (11) in term of U, χ, ξ and ζ. Only Wfoam in (8) are expressed in term of CD and u* which are related to U. Is it possible to express Wfoam by U instead of u*?

Response: It is possible but Equation will be rather bulky. We want to remain (8) as is for easy reading.

  1. Coefficient of determination (R2) is commonly used to show the performance of model predictions. It is suggested to show R2between the observations and the predictions.

Response: We calculated a coefficient of determination R2. Please see lines 274, 280, and 328.

Reviewer 2 Report

General comments:

1. Please add a review of the literature about wind waves modeling and measuring in the Gulf of Lion and  in the Gulf of Tehuantepec.

2. please add the Map of study area, big whole sea and Inset zoom map

3. Please add the statistical confirmation that the dependency is working on "Figure 6. Laws of development for sea surface fraction covered with foam  under limited wave". It may be confidence interval, R2 and others.

Minor comments;

1. Please add the work motivaton in the Absract 

2. In the abstract you say "fetches from 50 up to 7000"- in meters or other unit of measurement?

2. In the section 2 I see section 4.2, please please fix it.  The names of these sections (4.2 (2.2) and 2.3 imply a description of the measurement data, but  you have a description of the approach or method. please please fix it

3. Fig2,3 - very big titles and very small images. Please provide a bigger images, titlles not more 12 type size

Author Response

We thank the reviewers for their thoughtful comments on our manuscript. Following reviews, we made substantial modifications to improve the original manuscript. All of the reviewers’ comments have been taken into account.

Bellow original Reviewer’s comments are italicized, 10 pt., and our responses are written in “normal” print, 12 pt. 

Response to Reviewer #2 comments

  1. Please add a review of the literature about wind waves modeling and measuring in the Gulf of Lion and  in the Gulf of Tehuantepec.

Response: We added Figure 1 and a paragraph including 8 references (see lines 102-112) to describe areas of the Gulf of Lion and the Gulf of Tehuantepec. We refer to this literature throughout the entire manuscript, including the Introduction.

  1. please add the Map of study area, big whole sea and Inset zoom map

Response: Done, please see Figure 1.

  1. Please add the statistical confirmation that the dependency is working on "Figure 6. Laws of development for sea surface fraction covered with foam  under limited wave". It may be confidence interval, R2 and others.

Response: We calculated a coefficient of determination R2. Please see lines 274, 280, and 328.

Minor comments;

  1. Please add the work motivaton in the Absract 

Response: Done. We changed the first sentence in the Abstract:

“In this paper, we aimed to clarify the problem of foam coverage dependence on wave fetch, which is of interest in satellite microwave radiometry, but to which controversial results were reported previously.”

  1. In the abstract you say "fetches from 50 up to 7000"- in meters or other unit of measurement?

Response: We rewrite this sentence: “This development of foam coverage occurred at the range of dimensionless fetches from 50 up to 7000 if the fetch is scaled using wind velocity and gravity acceleration.”

  1. In the section 2 I see section 4.2, please please fix it. The names of these sections (4.2 (2.2) and 2.3 imply a description of the measurement data, but  you have a description of the approach or method. please please fix it

Response: We corrected the subsection number and titles.

  1. Fig2,3 - very big titles and very small images. Please provide a bigger images, titlles not more 12 type size

Response: Done, please see Figures 3 and 4.

Reviewer 3 Report

The paper presents an investigation of wave breaking using the Landsat-8 imagery. The topic is of great significance. However, it is not clear what is the new finding of this study. The innovation needs to be further clarified in the introduction part.

The study considered two scenes. One is during the Mistral storm winds (10:49 GMT March 5, 2015) in the Gulf of Lion, the Mediterranean Sea, and the other is during Tehuano storm winds (16:42 GMT January 13, 2015) in the Gulf of Tehuantepec, Pacific. Only one moment has been analyzed in each gulf. It would be better if more images at different times can be involved in the study.

As shown in Figure 5(b), there are only two wind speeds. If more images with different wind speeds can be involved, a more convincing result would be provided.

All the data in Figure 4 show a good agreement with the fitting line. But the trend of data in the Gulf of Lion (blue circles) seems slightly different with the overall gradient. Have you ever tried to fit the data of the Gulf of Lion separately? Can you explain the difference?  

As shown in Figure 7, there are large difference between this study and Kleiss & Melville (2010). Are the definition and evaluation method same with Kleiss and Melville (2010)? As mentioned in line 162, the surface fraction covered by wave breaking W was estimated in each pixel of the image. Please provide more information how to quantitatively evaluate the W, and then make a more fair and reasonable comparison.

It would be better to add two maps showing the water depth of the two study areas.

Some equations are not numbered, for example line 60 and 62. Some are only showing parameters, for example equation (3) in line 215, which should not stand alone in a row.

Author Response

We thank the reviewers for their thoughtful comments on our manuscript. Following reviews, we made substantial modifications to improve the original manuscript. All of the reviewers’ comments have been taken into account.

Bellow original Reviewer’s comments are italicized, 10 pt., and our responses are written in “normal” print, 12 pt. 

Response to Reviewer #3 comments

  1. The paper presents an investigation of wave breaking using the Landsat-8 imagery. The topic is of great significance. However, it is not clear what is the new finding of this study. The innovation needs to be further clarified in the introduction part.

Response: We aimed to clarify the foam coverage dependence on wave fetch, about which controversial results were reported previously. For this, we proposed a more correct experiment setting than previously used and performed such an experiment. To our mind, it is the main novelty of this work. Also, we obtained physically consistent results and proposed a model to explain them. We clarified this in lines 79-89:

“Among of the possible causes of disagreements are the unsteadiness of wave fields [6,55] and influence of currents and tides [68,57] during wavebreaking estimations. Wave development laws may be distorted for the unsteady wave field [6,7]. It reflects in wavebreaking intensity as reported in [55,57,68,75]. So comparing the wave-breaking estimations obtained in different time moments or experiments can lead to erroneous results if these effects are disregarded. However, if we perform wave measurements complemented with the wave breaking estimations simultaneously along the entire wave fetch, we can i) inspect whether wave characteristics obey known stationary fetch-limited laws; ii) if it does, consider wave breaking dependence on the fetch. Satellite data provide the opportunity to perform such experiments.”

  1. The study considered two scenes. One is during the Mistral storm winds (10:49 GMT March 5, 2015) in the Gulf of Lion, the Mediterranean Sea, and the other is during Tehuano storm winds (16:42 GMT January 13, 2015) in the Gulf of Tehuantepec, Pacific. Only one moment has been analyzed in each gulf. It would be better if more images at different times can be involved in the study.
  2. As shown in Figure 5(b), there are only two wind speeds. If more images with different wind speeds can be involved, a more convincing result would be provided.

Response to comments 2 and 3: You are right, of course. We also wanted it. But free of cloud images of high-wind areas are infrequent. We clarified it by adding lines 102-112:

“Figure 1 shows the QuikSCAT wind velocity field average from 2000 to 2009 years. Red rectangles in the figure show two near coastal areas, which are prominent by relatively high mean velocity caused by catabatic jets of Mistral and Tehuantepec winds. Waves generated by these winds develop from the shoreline in deep water conditions (see bathymetry of these areas in the figure). Wind wave investigations and wave modeling for these areas are presented in many papers (see, e.g., [100,101,102] for Mistral and [5,18,103] for Tehuantepec). So these places are rather suitable for investigations of wave development laws. However, scenes free of clouds are difficult to find using Landsat optical images for the study. We analyzed Landsat-8 images for several years but could select only two scenes, which positions in the Gulf of Lion and Gulf of Tehuantepec are shown in zoomed maps in Figure 1.”

  1. All the data in Figure 4 show a good agreement with the fitting line. But the trend of data in the Gulf of Lion (blue circles) seems slightly different with the overall gradient. Have you ever tried to fit the data of the Gulf of Lion separately? Can you explain the difference?  

Response: We cannot explain the difference. The wide spread of point clouds is typical for wave development dependencies in combining different data (see, e.g., [6,7], and literature cited there). We added separate estimation of our data trend, changed Figure 5 (former Figure 4), and rewrite this paragraph (lines 270-288):

“Least-square estimation applied to logarithms of our data yields power law

                   (3)

with a coefficient of determination . The joint cloud of all data in the figure leads to power law

                  (4)

with a coefficient of determination . Hereinafter, the confidence intervals correspond to double standard error [73]. Figure 5 shows both laws. Both estimations agree with generally accepted views on the values of parameters of the law (1) (see, e.g., [6,8,9]).

Thus, two selected scenes can be with certainty considered as wave development at limited fetches, and data on dimensionless fetch  can be used to study the law of whitecap coverage development. Further in this paper, we accept the law (4) as based on a more extensive dataset and haven better coefficient of determination.”

  1. As shown in Figure 7, there are large difference between this study and Kleiss & Melville (2010). Are the definition and evaluation method same with Kleiss and Melville (2010)? As mentioned in line 162, the surface fraction covered by wave breaking W was estimated in each pixel of the image. Please provide more information how to quantitatively evaluate the W, and then make a more fair and reasonable comparison.

Response: First, we added lines 193-196:

“These measurements differ from traditional wave-breaking estimations based on sophisticated analyses of optical sea images [18,19,77]. They are closer to radiometric estimations of foam coverage [50,60].“

Second, we extended Figure 8 (former Figure 7) and section Discussion (lines 383-397):

“Figure 8 summarizes the results of the discussion. First, it shows the uncertainty of known data regarding the dependency of wave breaking coverage on fetch and wave age. Data [18,55,58] are presented as examples. Bin-averaged data of Kleiss and Melville were taken from Figures 9b and 9d of [18]. Parameterizations of Lafon et al. [55, Equation (11)] and Callaghan et al. [57, Equation (3)] are

,

,

respectively. These results are obtained by combining data for different wind velocities, so they are independent of U, although such a dependence should be [60].”     

Second, Figures 8a and 8b show our parameterizations (9) and (11), respectively, in the ranges of parameters in which they are derived in this study. Note that disagreement in values with other data may be linked with different methods of wavebreaking measurements.”

  1. It would be better to add two maps showing the water depth of the two study areas.

Response: Done, see Figure 1. We also added lines 262-265:

“Classical laws (1) and (2) are valid for deep water only [1]. The waves are considered to be in deep water if half of the wavelength does not exceed the sea depth [3]. Figures 3c and 4c show 30m-isobaths to confirm that our wavelength measurements were in deep water areas.”

  1. Some equations are not numbered, for example line 60 and 62. Some are only showing parameters, for example equation (3) in line 215, which should not stand alone in a row.

Response: We deleted numbered line 215. We numbered only equations that are referred to in the text. As this numbering does not break the rules of this journal, we want to remain it for easy reading.

Reviewer 4 Report

This manuscript study whether sea foam grows with the increase of wave age, and presents a simple model of the foam coverage growth with wave fetch. I read your paper with lots of interest because the sea foam is important when using the electromagnetic scattering model to simulate the radar backscatter signal.

The paper is clear and informative.

Some minor comments follow:

1.       The data of the Scatterometer is greatly affected by the rainfall. Do you remove the wind field affected by the rainfall?

2.       Line 116, ‘Scaterrometry’ -> ‘Scatterometer’

3.       Considering the drawing specification, the unit of the coordinate axis needs to be marked on each figure, such as 'Longitude (deg)'

4.       Line 159, ‘…which agrees with other authors’ measurements’, here, references need to be given.

5.       Line 303, ‘и’ ?

6.       Line 348, ‘…14.4 и 21.3 m/s’, what does и mean?

7.       Line 126, ‘Figure 1 a’, there should be no spaces between 1 and a. There are many similar mistakes in the manuscript, please correct them.

Author Response

We thank the reviewers for their thoughtful comments on our manuscript. Following reviews, we made substantial modifications to improve the original manuscript. All of the reviewers’ comments have been taken into account.

Bellow original Reviewer’s comments are italicized, 10 pt., and our responses are written in “normal” print, 12 pt. 

Response to Reviewer #4 comments

  1. The data of the Scatterometer is greatly affected by the rainfall. Do you remove the wind field affected by the rainfall?

Response: Yes, we used the data of the Scatterometer for which rainfall flag is available. As follows from analyzed Landsat-8 data (see Figures 3 and 4), both scenes were without rainfall clouds.

  1. Line 116, ‘Scaterrometry’ -> ‘Scatterometer’

Response: Corrected

  1. Considering the drawing specification, the unit of the coordinate axis needs to be marked on each figure, such as 'Longitude (deg)'

Response: Done, see Figures 3 and 4

  1. Line 159, ‘…which agrees with other authors’ measurements’, here, references need to be given.

Response: Done, see Lines 184-185:

“which agrees with other authors’ measurements [20,48,68]. “

  1. Line 303, ‘и’ ?

Response: Corrected

  1. Line 348, ‘…14.4 и 21.3 m/s’, what does и mean?

Response: Corrected

  1. Line 126, ‘Figure 1 a’, there should be no spaces between 1 and a. There are many similar mistakes in the manuscript, please correct them.

Response: Corrected for all

Round 2

Reviewer 2 Report

Good work!

Reviewer 3 Report

The manuscript has been greatly improved. I have no further comment and suggest this paper can be published.